# VIA: Unified Spatiotemporal Video Adaptation for Global and Local Video Editing

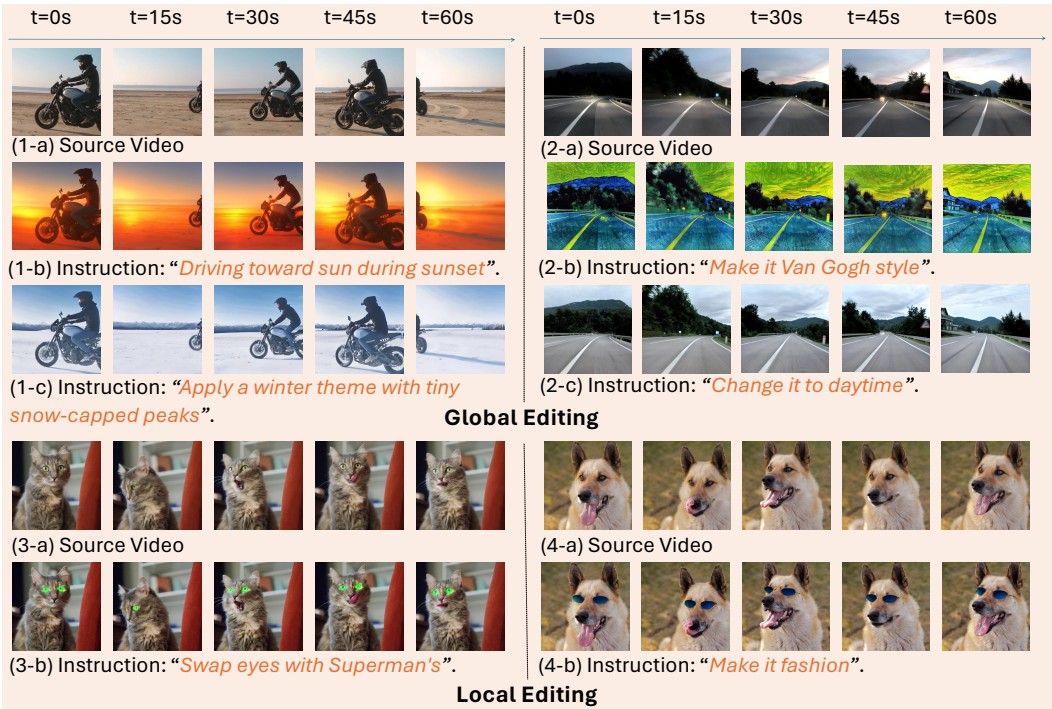

Figure 1: **Video editing results by VIA.** VIA excels in *precise* and *consistent* editing across diverse video editing tasks. Top: consistent results over long videos with a duration of 1 minute, which is challenging in current literature. Bottom: consistent results for precise local editing.

## ABSTRACT

Video editing is a cornerstone of digital media, from entertainment and education to professional communication. However, previous methods often overlook the necessity of comprehensively understanding both global and local contexts, leading to inaccurate and inconsistent edits in the spatiotemporal dimension, especially for long videos. In this paper, we introduce VIA, a unified spatiotemporal VIdeo Adaptation framework for global and local video editing, pushing the limits of consistently editing minute-long videos. First, to ensure local consistency within individual frames, we designed *test-time editing adaptation* to adapt a pre-trained image editing model for improving consistency between potential editing directions and the text instruction, and adapts masked latent variables for precise local control. Furthermore, to maintain global consistency over the video sequence, we introduce *spatiotemporal adaptation* that recursively *gather* consistent attention variables in key frames and strategically applies them across the whole sequence to realize the editing effects. Extensive experiments demonstrate that, compared to baseline methods, our VIA approach produces edits that are more faithful to the source videos, more coherent in the spatiotemporal context, and more precise in local control. More importantly, we show that VIA can achieve consistent long video editing in minutes, unlocking the potential for advanced video editing tasks over long video sequences.

## 1 INTRODUCTION

With the exponential growth of digital content creation, video editing has become essential across various domains, including filmmaking (Frierson, 2018; Dancyger, 2018), advertising (Mei et al., 2007; Kholisoh et al., 2021), education (Calandra et al., 2008; 2009), and social media (Jackson, 2016; Schmitz et al., 2006). This task presents significant challenges, such as preserving the integrity of the original video, accurately following user instructions, and ensuring consistent editing quality across both time and space. These challenges are particularly pronounced in longer videos, where maintaining long-range spatiotemporal consistency is critical.

A substantial body of research has explored video editing models. One approach uses video models to process the source video as a whole (Ku et al., 2024; Liu et al., 2023). However, due to limitations in model capacity and hardware, these methods are typically effective only for short videos (fewer than 200 frames). To overcome these limitations, various methods have been proposed (Xing et al., 2023; Wu et al., 2023; Guo et al., 2023; Wu et al., 2024). Another line of research leverages the success of image-based models (Ho & Salimans, 2022; Nichol et al., 2022; Podell et al., 2023; Avrahami et al., 2022; Brooks et al., 2023a) by adapting their image-editing capabilities to ensure temporal consistency during test time (Khachatryan et al., 2023; Geyer et al., 2024; Wu et al., 2024; Qi et al., 2023; Wang et al., 2023). However, inconsistencies accumulate in this frame-by-frame editing process, causing the edited video to deviate significantly from the original source over time. This accumulation of errors makes it challenging to maintain visual coherence and fidelity, especially in long videos. A significant gap remains in addressing both global and local contexts, leading to inaccuracies and inconsistencies across the spatiotemporal dimension. Current techniques often prioritize overall performance while neglecting the subtle aspects of consistency. This results in challenges when trying to preserve smooth transitions between frames and accurately execute edits, especially in longer or more intricate videos

To address these challenges, we introduce VIA, a unified spatiotemporal video adaptation framework designed for faithful, consistent, and precise video editing, pushing the boundaries of editing minute-long videos, as shown in Fig. 1. First, our framework introduces a novel *test-time editing adaptation* mechanism that adapts a pretrained image editing model to improve the semantic understanding of the source video and ensure consistency between editing directions and the text instructions. We propose an augmentation pipeline to create an in-domain tuning set for test-time adaptation, allowing the image editing model to learn associations between specific visual editing directions and corresponding instructions. This significantly enhances semantic comprehension and editing consistency within individual frames. To further improve local consistency, we introduce local latent adaptation to control local edits across frames, ensuring frame consistency before and after editing.

Second, effective editing requires seamless transitions and consistent edits, especially for long videos. To address this, we introduce *spatiotemporal attention adaptation* to maintain global editing coherence across the edited frames. Specifically, we propose *gather-and-swap* to *gather* consistent attention variables from the model's architecture and strategically apply them throughout the video sequence. This approach not only aligns with the continuity of the video but also reinforces the fidelity of the editing process, ensuring that changes are harmonized across frames over time.

Through rigorous testing and evaluation, our methods have demonstrated superior performance compared to existing techniques, delivering significant improvements in both local edit precision and the overall aesthetic quality of the videos. Moreover, our approach is considerably faster than previous methods due to the parallelized swapping process. To the best of our knowledge, we are the first to achieve consistent editing of minute-long videos. Our main contributions are as follows:

- **We introduce VIA, a novel framework designed to enable faithful, consistent, precise, and fast video editing.** Our approach pushes the boundaries of current video editing methods, ensuring both local and global consistency across the entire video.
- We introduce *spatiotemporal attention adaptation* to maintain global editing consistency across frames and proposed *gather-and-swap* to ensure coherent edits throughout the video.
- We propose a novel **test-time adaptation** mechanism that leverages an image editing model for video editing, enhancing the model's ability to follow text-based instructions and maintain semantic consistency within individual frames.

- **Our approach outperforms existing techniques in human evaluation and automatic evaluation**, delivering significantly better performance in terms of editing quality and efficiency.

## 2 RELATED WORK

### 2.1 TEXT-DRIVEN VIDEO EDITING

Text-driven Video Editing is a process to modify videos according to the instructions given by user. Inspired by the remarkable success of text-driven image editing (Avrahami et al., 2022; Brooks et al., 2023a; Tumanyan et al., 2023; Sheynin et al., 2023; Zhang et al., 2023), extensive methods have been proposed for video content editing (Qin et al., 2023; Khachatryan et al., 2023; Geyer et al., 2024; Wu et al., 2024; Qi et al., 2023; Wang et al., 2023; Ku et al., 2024). One paradigm for video editing is to adapt an image-based model to video. For example, Khachatryan et al. (2023) adapts image editing to the video domain without any training or fine-tuning by changing the self-attention mechanisms in Instruct-Pix2Pix to cross-frame attentions. Geyer et al. (2024) explicitly propagates diffusion features based on inter-frame correspondences to enforce consistency in the diffusion feature space. Yang et al. (2023b) construct a neural video field to enable encoding long videos with hundreds of frames in a memory-efficient manner and then update the video field with image-based model to impart text-driven editing effects. Ku et al. (2024) plug in any existing image editing tools to support an extensive array of video editing tasks. However, these methods are constrained by their ability to maintain global and local consistency, limiting to edit short videos within seconds. To efficiently enable longer video editing, Wu et al. (2024) centers on the concept of anchor-based cross-frame attention, firstly achieving editing 27 seconds videos. In our work, we built upon this line of work and improve editing and spatiotemporal consistency, firstly pushing the limits of video editing to minutes-long videos.

### 2.2 TEST-TIME ADAPTATION

Image-based video editing faces the challenge of ensuring temporal consistency during test time. To address this, Wu et al. (2023) propose to finetune a text-to-image model on a test video, enabling the generated videos with similar motion patterns to the source video. Xing et al. (2023) proposes light-weight spatial and temporal adapters for efficient one-shot video editing. Guo et al. (2023) adds a motion modeling module to the frozen based text-to-image model, and trains it on video clips, thereby distilling a reasonable motion prior. Wu et al. (2024) uses the same training set that was used to training the image editing model, and applies a data augmentation strategy for continuing pretraining to make the model equivariant to affine transformations. Different from the above approaches, we propose two orthogonal approaches that employs inference-time finetuing and local latent adaption, ensuring consistent and precise editing across frames.

### 2.3 SPATIOTEMPORAL CONSISTENCY

Ensuring spatiotemporal consistency is critical for video editing, especially for long videos. Qi et al. (2023) makes the attempt to study and utilize the cross-attention and spatial-temporal self-attention during DDIM inversion. Wang et al. (2023) proposes a spatial regularization module to fidelity to the original video. Park et al. (2024) presents spectral motion alignment (SMA), a framework that learns motion patterns by incorporating frequency-domain regularization, facilitating the learning of whole-frame global motion dynamics, and mitigating spatial artifacts. Ceylan et al. (2023) and Wu et al. (2023) improve the design of spatial attention to cross-frame attention to ensure consistency. In our work, we further ensure consistency inside the anchor-based frames and propose a two-step gather-swap process to adapt spatiotemporal attention for consistent global editing.

## 3 PRELIMINARIES

**Diffusion Models.** In this work, we adapt an image editing model for instruction-based video editing. Given an image $x$, the diffusion process produces a noisy latent $z_t$ from the encoded latent $z = \mathcal{E}(x)$ where the noise level increases over timesteps $t \in T$. A network $\epsilon_\theta$ is trained to minimize

the following optimization problem,

$$\min_\theta \mathbb{E}_{y,\epsilon,t}\left[\|\epsilon - \epsilon_\theta(z_t, t, \mathcal{E}(c_I), c_T)\|\right] \tag{1}$$

where $\epsilon \in \mathcal{N}(0,1)$ is the noise added by the diffusion process and $y = (c_T, c_I, x)$ is a triplet of instruction, input image and target image. Here $\epsilon_\theta$ uses a U-Net architecture (Ronneberger et al., 2015), including convolutional blocks, as well as self-attention and cross-attention layers.

**Attention Layer.** The attention layer first computes the attention map using query, $\mathbf{Q} \in \mathbb{R}^{n_q \times d}$, and key, $\mathbf{K} \in \mathbb{R}^{n_k \times d}$ where $d$, $n_q$ and $n_k$ are the hidden dimension and the numbers of the query and key tokens respectively. Then, the attention map is applied to the value, $\mathbf{V} \in \mathbb{R}^{n \times d}$ as follows:

$$\mathbf{Z}' = \text{Attention}(\mathbf{Q}, \mathbf{K}, \mathbf{V}) = \text{Softmax}(\frac{\mathbf{Q}\mathbf{K}^\top}{\sqrt{d}})\mathbf{V}, \tag{2}$$

$$\mathbf{Q} = \mathbf{Z}\mathbf{W}_q, \quad \mathbf{K} = \mathbf{C}\mathbf{W}_k, \quad \mathbf{V} = \mathbf{C}\mathbf{W}_v, \tag{3}$$

where $\mathbf{W}_q, \mathbf{W}_k, \mathbf{W}_v$ are the projection matrices to map the different inputs to the same hidden dimension $d$. $\mathbf{Z}$ is the hidden state and $\mathbf{C}$ is the condition. For self attention layers, the condition is the hidden state while the condition is text conditioning in cross attention layers.

**Cross-frame Attention.** Given $N$ frames from the source video, cross-frame attention has been employed in video editing by incorporating $\mathbf{K}$ and $\mathbf{V}$ from previous frames into the current frame's editing process (Liu et al., 2023; Wang et al., 2023; Wu et al., 2024), as shown below:

$$\phi = \text{Softmax}\left(\frac{\mathbf{Q}_{\text{curr}}[\mathbf{K}_{\text{curr}}, \mathbf{K}_{\text{group}}]^{\mathbf{T}}}{\sqrt{d}}\right)[\mathbf{V}_{\text{curr}}, \mathbf{V}_{\text{group}}], \tag{4}$$

where $\mathbf{K}_{\text{group}} = [\mathbf{K}^0, \ldots, \mathbf{K}^k]$ and $\mathbf{V}_{\text{group}} = [\mathbf{V}^0, \ldots, \mathbf{V}^k]$, and $k$ is the group size. By incorporating $\mathbf{K}_{\text{group}}$ and $\mathbf{V}_{\text{group}}$ during the video editing process for each frame, the temporal consistency is improved. In this paper, we improve cross-frame attention with a two stage gather-swap process to significantly improve the spatiotemporal consistency.

# 4 THE VIA FRAMEWORK

We introduce a unified framework to tackle key challenges in instruction-guided video editing, with a focus on ensuring editing consistency and spatiotemporal coherence across video frames by leveraging an image editing model, as shown in Fig. 3. Below, we outline the distinct methodologies that form the foundation of our approach.

## 4.1 TEST-TIME EDITING ADAPTATION FOR CONSISTENT LOCAL EDITING

Videos often exhibit substantial variation across the temporal dimension, particularly in long sequences, making it crucial for the model to maintain consistency in the editing process for each frame. When adapting image editing models for video editing, the same instructions must yield consistent semantic interpretations across frames—for example, every frame should exhibit the same degree of darkness when instructed to *"make it night."* Additionally, non-target elements in each frame must remain unchanged; for instance, a table should remain intact when the instruction is to replace an apple with an orange. To address these challenges, we propose two orthogonal approaches to achieve consistent local editing.

Inspired by DreamBooth (Ruiz et al., 2023), which employs inference-time fine-tuning to associate specific objects with unique textual tokens, we similarly link visual editing outcomes with corresponding instructions, as shown in Fig. 2. We begin with a pipeline to generate the in-domain tuning set without the need for external resources. For the video to be edited, the image editing model $\Psi$ first edits a randomly sampled frame $S_{\text{root}}$ to get editing result $E_{\text{root}}$. Then we apply random affine transformations to both the edited frame and source frame. Consider $\mathcal{F}_k$ as affine transformation:

$$T = \{(\mathcal{F}_k(S), \mathcal{F}_k(E), I) \mid \mathcal{F}_k \in \mathcal{F}\} \tag{5}$$

where $\mathcal{F}$ is the set of transformations. The tuning set $T$ consists of triples: source image, edited image, and editing instruction. By fine-tuning the image editing model $\Psi$ on this domain-specific

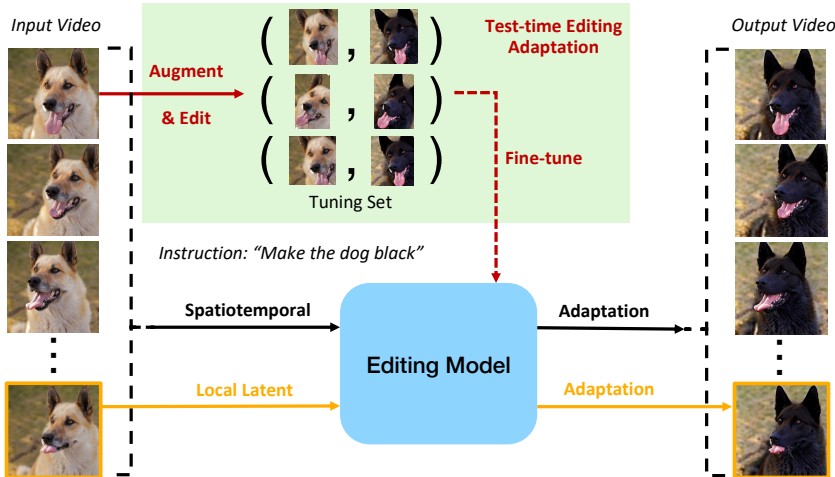

Figure 2: **Overview of our VIA framework.** For local consistency, Test-time Editing Adaptation finetunes the editing model with augmented editing pairs to ensure consistent editing directions with the text instruction, and Local Latent Adaptation achieves precise editing control and preserves non-target pixels from the input video. For global consistency, Spatiotemporal Adaptation collects and applies key attention variables across all frames.

dataset, the model learns to map specific visual editing directions to the corresponding instructions. It enhances semantic consistency across the video, particularly for instructions that lack detailed editing specifications, by reducing divergent editing outcomes across different frames.

For the second challenge, where editing instructions specify alterations to specific areas, current video editing models often unintentionally modify regions that the user did not target. To resolve this, we propose **Progressive Boundary Integration** during the sampling process. It integrates the inverted latent representation with the generated latent at each timestep, ensuring that modifications remain confined to the designated areas while preserving the integrity of non-targeted regions. Our approach compels the model to strictly adhere to the editing instructions, focusing exclusively on the specified areas. Unlike in the image domain, where background preservation is achieved through localized edits by blending latent representations $z$ from the source and target images (Cao et al., 2023; Gu et al., 2024), our approach smoothly merges source and target latents via linear interpolation between 0 and 1 over the time series. The mathematical representation is given by:

$$\mathbf{M}_{\mathrm{src}}(x,y) = \begin{cases} \mathbf{M}_{\mathrm{src}}(x,y) \cdot \frac{t}{T}, & \text{if } t \leq T \text{ and } \mathbf{M}_{\mathrm{src}}(x,y) = 1 \\ \mathbf{M}_{\mathrm{src}}(x,y), & \text{otherwise} \end{cases} \tag{6}$$

Here, $\mathbf{M}_{\mathrm{src}}(x,y)$ is predefined as 1 in a specific central area and 0 elsewhere. Within this central area, $\mathbf{M}_{\mathrm{src}}(x,y)$ incrementally increases from 0 to 1 over $T$ steps, while the values outside this central region remain unchanged. By applying these masks to define the editing region, VIA was able to achieve precise and targeted editing. To facilitate large-scale video editing, we have also implemented an automatic mask generation process, which is described in detail in the Appendix.

## 4.2 SPATIOTEMPORAL ADAPTATION FOR CONSISTENT GLOBAL EDITING

For long video editing, maintaining smooth transitions without glitches or artifacts is essential. Attention variables within the U-net have been found to strongly correlate with the generated content. To ensure consistent global editing, we propose a two-step *gather-and-swap* process to adapt spatiotemporal attention, as illustrated in Fig. 3. In this method, the gathered attention group is uniformly applied across all frames, ensuring internal coherence throughout the editing process and preventing inconsistencies in the edited video.

Firstly, in the *gather* stage, the model progressively edits the image, with key $\mathbf{K}$ and value $\mathbf{V}$ from previous frames in the group, rather from their own $\mathbf{K}_{\mathrm{curr}}$ and $\mathbf{V}_{\mathrm{curr}}$,

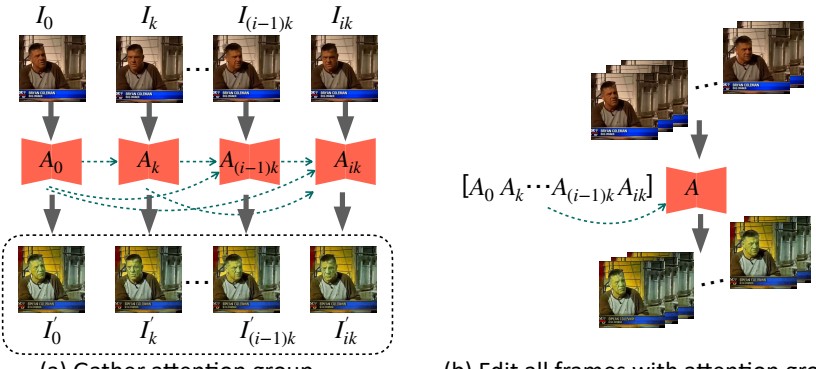

(a) Gather attention group          (b) Edit all frames with attention group

Figure 3: **The *gather-and-swap* process for video editing.** The left part of the diagram illustrates the gathering process. We initially sample $k + 1$ frames evenly distributed throughout the video. The first frame undergoes standard editing using an image editing model, during which the attention variables are captured and stored. For each of the subsequent $k$ frames, the attention variable from the preceding frame is swapped in, and its own attention variables are also preserved. In the right part, the collected attention variables from all $k + 1$ frames are swapped into the editing process of each frame. This includes applying the previously gathered attention variables to enhance the consistency and quality of edits across the sequence.

$$\phi = \text{softmax}\left(\frac{\mathbf{Q}_{\text{curr}}\mathbf{K}_{\text{prev}}^{T}}{\sqrt{d}}\right)\mathbf{V}_{\text{prev}},\tag{7}$$

$$\mathbf{K}_{\text{group}}^{(t+1)} = [\mathbf{K}_{\text{group}}^{(t)}, \mathbf{K}_{\text{curr}}], \quad \mathbf{V}_{\text{group}}^{(t+1)} = [\mathbf{V}_{\text{group}}^{(t)}, \mathbf{V}_{\text{curr}}]\tag{8}$$

Since $\mathbf{K}_{\text{curr}}$ and $\mathbf{V}_{\text{curr}}$ are calculated by the $\phi$ from the last layer, which already has a stronger dependency on other frames, the saved elements have a stronger consistency with previous group elements, leading to in-group consistency in $\mathbf{K}_{\text{group}}^{(k+1)}$ and $\mathbf{V}_{\text{group}}^{(k+1)}$.

In the second stage, we apply the attention group to the editing process of all frames, including those originally used to generate the attention group. This approach resolves the inconsistency in the first few frames, where they initially have less dependency on other frames. Throughout the editing process, each frame continues to refrain from using its own attention variables, instead relying on the shared attention group to maintain consistency across the entire video. This ensures that all frames, even the earlier ones, are edited with a global perspective, reducing discrepancies between frames.

$$\phi = \text{softmax}\left(\frac{\mathbf{Q}_{\text{curr}}\mathbf{K}_{\text{group}}^{T}}{\sqrt{d}}\right)\mathbf{V}_{\text{group}},\tag{9}$$

In this way, all frames share the same attention group, which is internally consistent, leading to maximum coherence between the edited frames. The *swap* process is distributed across multiple GPUs, enabling parallel frame editing, which significantly reduces editing time. Moreover, while previous work has primarily relied on self-attention for cross-frame consistency, we discovered that cross-attention also plays a crucial role in maintaining coherence. Combining both self-attention and cross-attention mechanisms yields the most effective editing outcomes. To further enhance the process, we select attention variables from frames that are evenly distributed throughout the video. This ensures comprehensive coverage of the dynamic changes across the video, capturing a broad representation of frame differences and maximizing consistency in the edits. Fig. 3 illustrates the two stages, where $\mathbf{A}$ represents both $\mathbf{K}$ and $\mathbf{V}$.

## 5   EVALUATION

In this paper, we adapt the open-source image editing model MGIE (Fu et al., 2024) for video editing. For spatiotemporal adaptation, we collect attention variables from four frames. To enhance the model's editing capabilities, we introduce the following transformations for each image pair,

aimed at increasing variability while maintaining the structural integrity of the images: (*i*) slight rotation (up to ±5 degrees); (*ii*) translation (up to 5% both horizontally and vertically); and (*iii*) after applying these transformations, cropping the images to between 75% and 100% of their original size to simulate changes in video sequence framing. Additionally, we apply shearing transformations of up to 10 degrees. These affine transformations introduce realistic variations, simulating the diversity of viewing angles typically encountered across different frames in a video. This approach helps the model adapt to the natural changes in perspective that occur during video sequences.

Table 1: **Human evaluation results.** We compare our model with five previous open-source methods from three aspects. 'Tie' indicates the two models are on par with each other. Only spatiotemporal adaptation is used when comparison with baseline models.

| | Ours | Rerender | Tie | Ours | TokenFlow | Tie | Ours | AnyV2V | Tie | Ours | Video-P2P | Tie | Ours | Tune-A-Video | Tie |
|---|---|---|---|---|---|---|---|---|---|---|---|---|---|---|---|
| Instruction Following | **50.50** | 34.00 | 15.5 | **75.75** | 16.00 | 8.25 | **56.00** | 29.00 | 15.00 | **74.00** | 16.25 | 9.75 | **70.25** | 20.75 | 9.00 |
| Consistency | **47.25** | 35.00 | 17.75 | **38.00** | 31.50 | 30.5 | **53.50** | 23.25 | 23.25 | **80.50** | 9.50 | 10.00 | **68.75** | 20.75 | 10.5 |
| Overall Quality | **53.50** | 29.00 | 17.5 | **61.75** | 22.75 | 15.5 | **63.50** | 30.00 | 6.5 | **63.75** | 22.75 | 13.5 | **56.00** | 22.25 | 21.75 |

For a comprehensive evaluation against state-of-the-art methods, we begin by comparing our results with the closed-source method Fairy (Wu et al., 2024), which is capable of handling videos up to 27 seconds in length. We use the same video from their paper to ensure a direct comparison. Additionally, we conduct both qualitative and human evaluations against open-source state-of-the-art baselines, including AnyV2V (Ku et al., 2024), Rerender (Yang et al., 2023a), Tokenflow (Geyer et al., 2024), Video-P2P (Liu et al., 2023), and Tune-A-Video (Wu et al., 2023). For the comparison with AnyV2V, we use the first edited frame generated by VIA as the starting point for the evaluation.

## 5.1 QUANTITATIVE EVALUATION

**Human Evaluation.** We began by conducting a human evaluation. Since many baselines are unable to handle long videos, we limited the video length to 4–8 seconds to ensure a fair comparison. All videos were standardized to a frame size of 512x512 pixels. A total of 400 videos were sampled for human evaluation to compare the performance of our VIA (Ours) against open-source state-of-the-art baselines, including Rerender, TokenFlow, AnyV2V, Video-P2P, and Tune-A-Video.

The evaluation was based on the following criteria: Instruction Following assesses how accurately the system follows user commands and instructions during the editing process, measuring its ability to execute specified edits as intended; Consistency measures the internal coherence of edits across frames, ensuring that transitions and edits maintain a consistent visual style and context throughout the video, avoiding abrupt changes or visual discrepancies. Overall Quality evaluates the final video's visual appeal and professional finish, considering factors such as clarity, smoothness, and the aesthetic quality of the edited video. These criteria were chosen to provide a comprehensive assessment of our system's performance, addressing both functional accuracy and the overall visual quality of the edited videos. The results, presented in Tab. 1, highlight the strengths of our proposed method, across Instruction Following and Consistency, where VIA performed exceptionally well. The overall performance also demonstrates the robustness of our approach over all other baselines.

**Automatic Evaluation.** We also conducted automatic evaluation as in Tab. 2. Frame-Acc (Qi et al., 2023; Yang et al., 2023a) measures the percentage of frames where the edited image has a higher CLIP similarity to the target prompt than the source prompt; Tem-Con (Esser et al., 2023) measures the temporal consistency via computing the cosine similarity between all pairs of consecutive frames. Pixel-MSE (Ceylan et al., 2023) is the average mean-squared pixel error between each frame and its corresponding target frame. VIA outperformed all other models across these metrics, delivering superior accuracy and consistency while also achieving faster processing speeds. We did not use test-time adaptation for VIA, as some of the baseline models do not inherently benefit from it, which ensured a fair comparison. Additionally, we calculated the evaluation latency of the editing process, which was carried out on an A100 machine with 8 GPUs. The global adaptation process could be distributed across multiple GPUs to further accelerate the process. Detailed speed analysis can be found in the Appendix.

Table 2: **Automatic evaluation results.** VIA outperforms open-sourced video editing models in automatic metrics. Only spatiotemporal adaptation is used when compared with baseline models.

|  | VIA | Rerender | TokenFlow | AnyV2V | Video-P2P | Tune-A-Video |
|---|---|---|---|---|---|---|
| Frame-Acc ↑ | **0.869** | 0.734 | 0.587 | 0.533 | 0.587 | 0.601 |
| Tem-Con ↑ | **0.983** | 0.954 | 0.932 | 0.856 | 0.912 | 0.927 |
| Pixel-MSE ↓ | **0.011** | 0.016 | 0.018 | 0.026 | 0.020 | 0.019 |
| Latency(sec) ↓ | **16** | 406 | 450 | 570 | 612 | 529 |

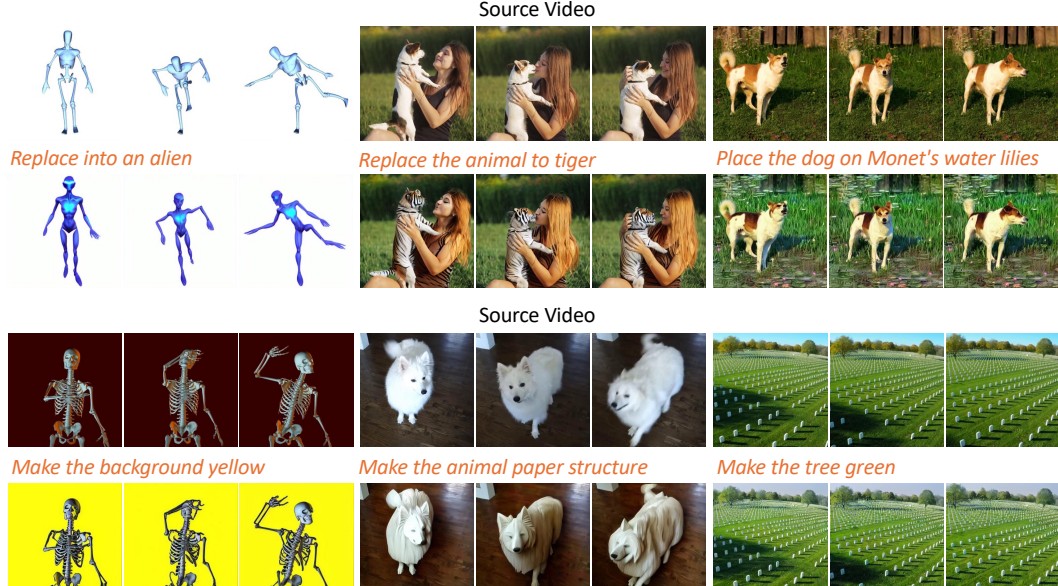

Figure 4: **Local editing results.** VIA is capable of performing a wide range of localized editing tasks, where only specific regions or pixels within a frame are modified. These tasks include identity swapping, object part editing, and background editing. The left column shows the outcomes of these editing operations applied to two 1-minute long videos.

## 5.2 QUALITATIVE RESULTS

**Local Editing Results.** Fig. 4 showcases the performance of VIA on various local editing tasks, where only specific parts of the frame are modified. VIA excels at accurately identifying the target area and applying precise edits, even in cases with occluded subjects, as demonstrated in the "Replace the animal with a tiger" example. Beyond foreground modifications, VIA performs exceptionally well in background edits. For example, it successfully "Places the dog on Monet's water lilies" in a video, seamlessly blending the subject into the new background. In the more challenging "skeleton video", where the background needs to fill gaps between the bones, VIA maintains consistent performance, ensuring that the dancing skeleton remains unaffected. Additional challenging tasks, such as local stylization, are detailed in the Appendix.

**Global Editing Results.** Fig. 5 highlights the global editing capabilities of VIA across a range of videos, demonstrating its ability to apply consistent, high-quality edits. A uniform set of editing instructions was used across different videos, resulting in coherent and visually appealing modifications throughout. The bottom example specifically illustrates VIA's proficiency in understanding and consistently applying visual effects across all frames, ensuring seamless transitions and maintaining the integrity of the visual narrative across the entire video.

**Long Video Editing.** A direct consequence of the high consistency feature in our video editing framework is its proficiency in handling longer videos, as shown in Fig. 1. Additional results on local and global editing are presented in Fig. 4 and Fig. 5, respectively. Currently, no existing video editing models are capable of editing minute-long videos due to limitations in their architectural design. Consequently, it is not possible to apply or compare our method with others on such long

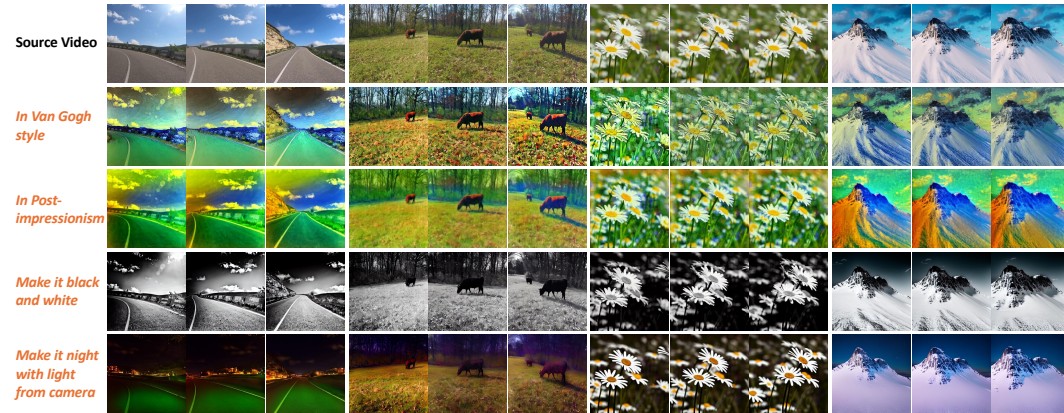

Figure 5: **Global editing results.** VIA demonstrates robust global editing performance across various videos using a consistent set of editing instructions, producing high-quality results. The left two columns are a 2-minute video and a 1-minute video.

videos. One of our baselines, Fairy (Wu et al., 2024), has not made their code publicly available, but they report that their model supports videos up to 27 seconds in length. We compare our results on the same video using identical editing instructions, as shown in Fig. 6. Notably, VIA demonstrates superior global and local consistency, which can be attributed to our unified adaptation framework.

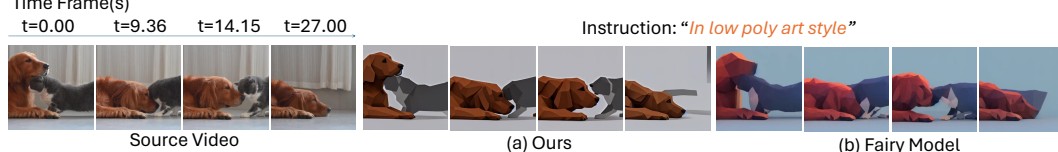

Figure 6: **Comparison with the baseline model on the long video.** We present the editing results on sampled frames from a 27-second duration video.

**Qualitative Comparison.** In Fig. 7, we present two examples of video editing to showcase the performance of VIA in comparison to other models. In the first example, the video depicts rapidly moving clouds against a blue sky, with the editing instruction to "Set the time to sunset." This task challenges the model to infer the necessary visual changes, such as adjusting lighting and color tones. Despite the swift movement of the clouds, which places a high demand on temporal consistency, VIA demonstrates excellent coherence across frames. The Editing Adaptation process allows VIA to effectively align the visual effect with the concept of "sunset," ensuring smooth and realistic changes. In contrast, other models struggled to execute the command adequately. Notably, the AnyV2V model partially achieved the desired visual effect by leveraging the initial frame generated by VIA. On the right, we show an object-swapping example where a monkey moves from within the frame to outside of it. The challenge here lies in maintaining a smooth transition from the full subject to a partially visible one, ensuring consistency in identity. While other methods often introduce artifacts and inconsistencies between the edited frames and the original video, VIA seamlessly swaps the subject's identity, preserving visual coherence and continuity throughout the transition.

From this comparison, we found that (1) VIA outperforms the baselines in both editing quality and processing speed. It ensures smooth transitions in edited videos, even when dealing with rapidly moving objects, while some models, such as AnyV2V, generate noticeable artifacts. (2) VIA demonstrates strong performance in adhering to complex instructions, where other models often struggle. While competing methods experience degraded performance with intricate commands, VIA consistently follows the instructions, applying edits accurately across all frames.

**Ablation on Individual Components.** In Fig. 8, we analyze the impact of various components of VIA on the editing of long videos. Our experiments indicate that the quality of the initial edited frames plays a critical role in determining the overall visual quality, as information from these root frames propagates throughout the video sequence. Test-time adaptation further enhances the model's

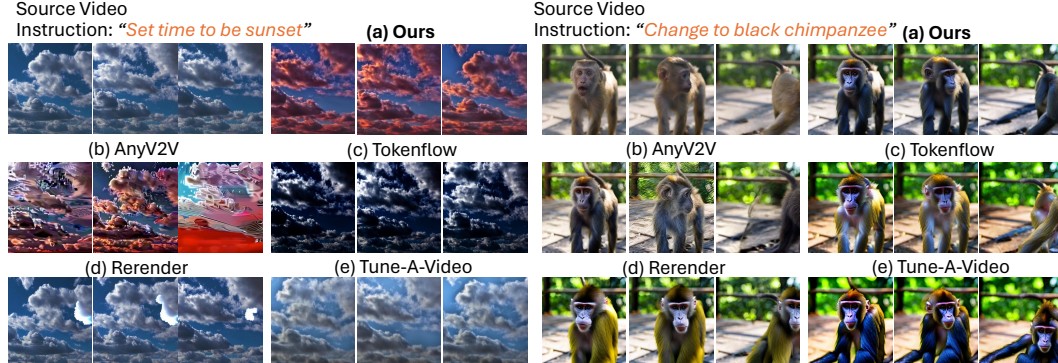

Figure 7: **Qualitative comparison with baselines.** VIA is able to produce consistent editing results.

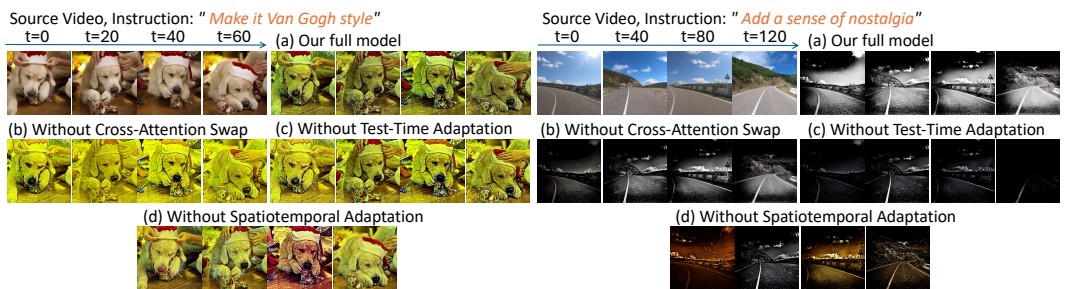

Figure 8: **Ablation Study on components in VIA on long video.** On the left, we present an example of 60 seconds video editing of stylization. On the right, we show video editing of 120 seconds. Test-time adaptation ensures robust visual effects that adhere to the given instructions. Without the gather-swap technique, object consistency across different frames is compromised. Furthermore, incorporating cross-attention, in addition to self-attention, enhances consistency and reduces artifacts.

ability to closely follow the editing instructions, improving overall consistency. When *gather-and-swap* is omitted and the model relies solely on cross-frame attention, inconsistencies start to emerge between frames. Additionally, although self-attention is commonly employed to ensure frame-to-frame consistency, we found that the inclusion of cross-attention significantly improves the quality of video editing. For example, in the left example, the omission of cross-attention results in variation in the hat color across frames. The combination of both attention mechanisms helps maintain uniformity in appearance and color, ensuring higher editing precision. For additional ablation studies, please refer to Appendix C.

## 6 LIMITATION

While VIA has demonstrated impressive performance in video editing, it is not without limitations. Firstly, it inherits constraints from the underlying image editing model, which restricts the range of editing tasks to those predefined by the image model. Secondly, although VIA performs well across a wide array of video editing tasks, its performance decreases when dealing with videos featuring complex interactions between objects. In the future, we plan to explore a more detailed part-to-part alignment to improve the model's capability in handling such scenarios.

## 7 CONCLUSION

This paper introduces a novel video editing framework that tackles the critical challenges of achieving temporal consistency and precise local edits. Our approach surpasses the limitations of traditional frame-by-frame methods, delivering coherent and immersive video experiences. Extensive experiments show that our framework outperforms existing baselines in terms of handling temporal dynamics, ensuring local edit precision, and enhancing overall video aesthetic quality. This advancement paves the way for new possibilities in media production and creative content generation, setting a new benchmark for future developments in video editing technology.

## ETHICS STATEMENT

This research adheres to ethical guidelines and practices in the development and application of video editing technologies. Our work focuses on improving the efficiency and quality of automated video editing, with the intent of advancing creative tools for legitimate purposes such as filmmaking, education, and advertising. We are mindful of the potential misuse of video editing technology, particularly in generating misleading or harmful content. To mitigate such risks, we strongly advocate for responsible use and encourage the implementation of safeguards to prevent misuse. Additionally, the data used in this research are publicly available and were utilized in compliance with all relevant legal and ethical standards. No personal or sensitive information was involved in the study.

## REPRODUCIBILITY STATEMENT

We are committed to ensuring the reproducibility of our research. All experimental details, including the model architecture, hyperparameter settings, and evaluation protocols, are thoroughly described in the paper. To facilitate replication, we provide access to the source code for spatiotemporal adaptation process in the supplementary material, which is used for comparison with baselines.

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

## A    ADDITIONAL IMPLEMENTATION DETAILS

The evaluation was conducted using a collection of online resources and video clips from Panda-70M Chen et al. (2024). VIA can be applied to general image editing frameworks Hertz et al. (2022); Brooks et al. (2023b); Fu et al. (2024). In this work, we used MGIE Fu et al. (2024) as the base image editing model. We set the diffusion step $T$ to 10 and performed spatiotemporal adaptation through all cross-attention and self-attention layers. Our experiments showed that adaptation achieves the best performance when conducted on at least the first 8 steps.

We also observed that increasing the total diffusion step $T$ improves image detail but simultaneously raises the probability of artifacts. Through experimentation, we found that using a value between 5 and 10 generally yields good editing results while maintaining high processing speed. This balance ensures high-quality edits without introducing undesirable visual inconsistencies.

## B    SPEED ANALYSIS

VIA not only achieves great performance, but also offers impressive speed. The fine-tuning process takes approximately 1 minute, regardless of the video's length. For the global adaptation process, it takes instructPix2Pix (Brooks et al., 2023a) about 1 second per frame, and MGIE (Fu et al., 2024) around 3 seconds per frame.

**Distribution Across GPUs:** Once we gather the frames, the editing for all frames can be performed on different GPUs simultaneously, as the frame editing process only depends on the fixed group frames. We utilize 8 GPUs for processing, which helps manage the load effectively.

**Total Processing Time for a 600-frame video:**

- **MGIE:** 60 (fine-tuning) + $\frac{3 \times 600}{8} = 285$ seconds.
- **InstructPix2Pix:** 60 (fine-tuning) + $\frac{1 \times 600}{8} = 135$ seconds.

For the comparison with baselines, where only spatio-temporal adaptation is used (without fine-tuning or local adaptation), the time is:

- **MGIE (without fine-tuning):** $\frac{3 \times 600}{8} = 225$ seconds.
- **InstructPix2Pix (without fine-tuning):** $\frac{1 \times 600}{8} = 75$ seconds.

## C    MORE ABLATION STUDY

In the main paper, we presented an ablation study on long videos. Here, we demonstrate the impact of various components of VIA on videos less than 20 seconds in duration, where a dog rapidly moves its head and shakes its body. The provided editing instruction was "Change into a tiger." Our Local Latent Adaptation process effectively identifies the target area and performs precise edits. Our experiments also reveal that the initial edited frames largely determine the overall visual quality, as information from these root frames propagates throughout the entire video sequence. Test-time adaptation further ensures that the model adheres closely to the editing instructions.

In the absence of the *gather-and-swap* process, relying solely on cross-frame attention results in inconsistencies across frames. Furthermore, while self-attention is commonly used to maintain frame consistency, we found that cross-attention significantly improves the quality of video editing. For example, when cross-attention is excluded, facial alignment with the source video is reduced, leading to less accurate transformations. In the right part of the experiment, we applied a style change to the video, transforming it into the aesthetic of a Japanese woodblock print.

## D    ANALYSIS ON FAILURE CASES

We highlight several failure cases where VIA did not achieve the expected performance, as shown in Fig. 10. The first challenge involves handling complex interactions. In the example on the left, while we successfully captured the intricate body dynamics during a sophisticated dance sequence, a

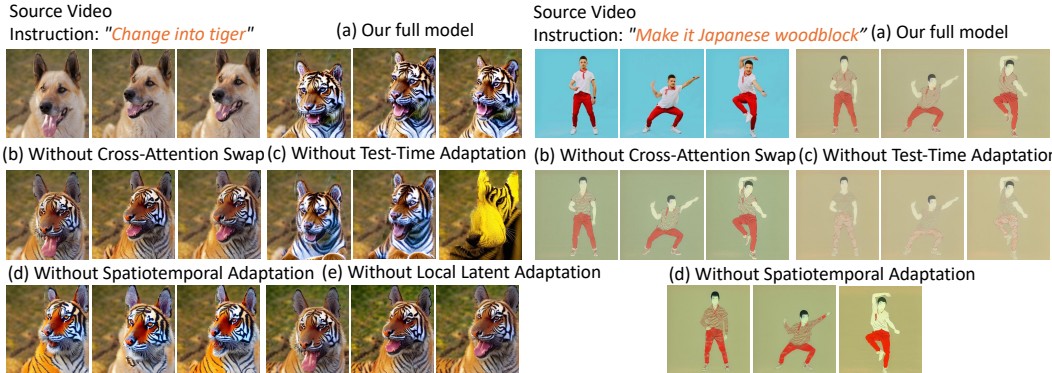

Figure 9: **Ablation study on videos less than 20 seconds.**

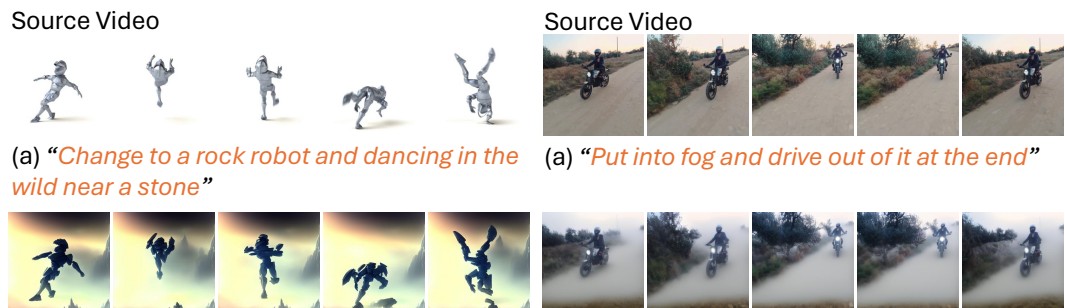

Figure 10: **Failure cases.** In the left example, a misalignment occurs during the interaction between the robot and the rock, despite accurately capturing the dance sequence. In the right example, while the driver is seamlessly integrated into the fog, the sequence fails to depict driving out process, leaving the edit incomplete.

misalignment occurred when the robot was supposed to interact with a rock, leading to inaccuracies at the point of contact. The second challenge relates to temporal dynamics. Although we seamlessly integrated the driver into the fog, the sequence did not show the car emerging from the fog, leaving the scene incomplete. In the future, we plan to incorporate more explicit temporal information into the editing process to better address these issues.

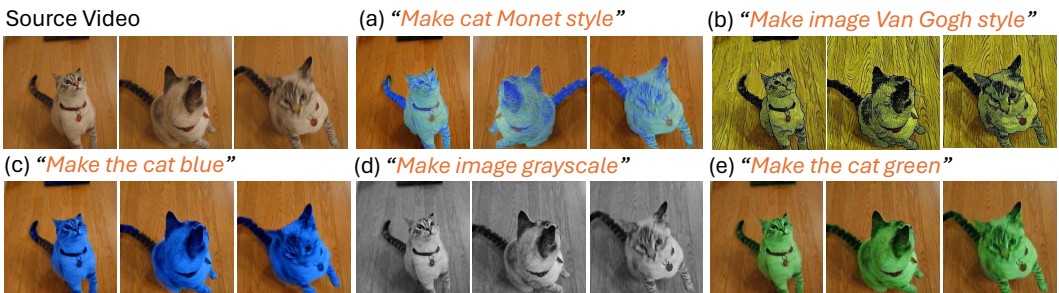

Figure 11: **Global and local stylization.** We show video editing results with different given instructions in (a)-(e). Local Editing in VIA is not limited to object swapping. Whereas other methods can only do stylization on the whole image, our model could achieve a local stylization.

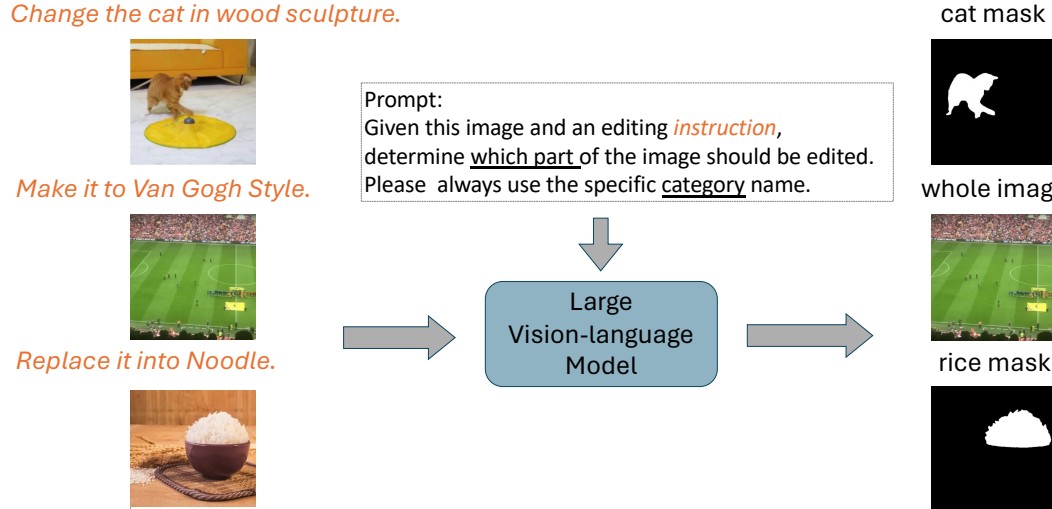

Figure 12: **Automatic mask generation.** A single frame from the video, along with a tailored text prompt encapsulating the editing instruction, is fed into a Large Vision-Language Model (LVLM), such as GPT-4, to generate a text description that specifies the region to be edited. If the designated editing area does not cover the entire image, this text description is then passed into a segmentation model, such as the Segment Anything model, to create a mask for the targeted region. This automated process allows for precise identification of the area to be modified, ensuring that only the relevant portion of the image is edited, while preserving the integrity of the rest of the frame.

## E  LOCAL STYLIZATION

Fig. 11 demonstrates the advanced video editing capabilities of our method, showcasing its ability to perform both global and local stylization. Unlike previous methods that are restricted to applying stylistic changes across the entire image, our approach enables precise, localized edits. This flexibility is illustrated through various examples in subfigures (a)-(e), where different instructions are applied to achieve distinct editing effects. Whether performing object swapping or applying regional stylization, our model overcomes the limitations of traditional methods by enabling targeted modifications while preserving the overall composition and aesthetic integrity of the video. This allows for greater control and precision in video editing, significantly enhancing creative possibilities.

## F  AUTOMATIC MASK GENERATION

We present an automated mask generation pipeline aimed at enhancing user experience and streamlining the editing process, particularly for large-scale edits. Editing instructions often specify modifications to specific regions, but current end-to-end models tend to alter unintended areas. To address this, we designed an automated pipeline for mask generation, as illustrated in Fig. 12.

First, a Large Vision-Language Model (GPT-4V in our experiment) is prompted to generate a textual description, $P$, of the region to be modified for each frame. Using this description, we apply the Segment Anything model (Kirillov et al., 2023) to extract a mask that accurately delineates the target area for editing. It is important to note that we did not use GPT-4V during comparisons with baselines in the original paper.

In the optimal setting, VIA involves further tuning in the local adaptation process, which some baselines do not utilize. For fairness in comparisons, we degraded our model to use only Spatiotemporal Adaptation during all evaluations. This ensures that our results are directly comparable to baseline models without additional enhancements from local adaptation or the automated mask generation process.

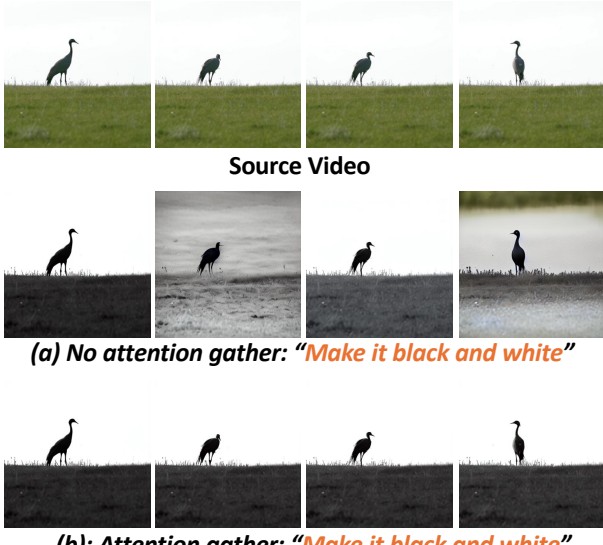

**Figure 13: The edited group frames with&without attention gathering process.** The gathering process ensures in-group consistency, providing a fixed visual editing direction for all frames.

## G   COMPARISON ON ATTENTION SWAPPING PROCESS

Attention variables within the U-net of diffusion models have proven to be highly correlated with the generated visual content and are widely used in various editing tasks (Hertz et al., 2022; Cao et al., 2023; Gu et al., 2023; Liu et al., 2023; Ceylan et al., 2023). In video editing, some methods train models to reconstruct the original videos and swap key attention features during the editing process (Ku et al., 2024; Liu et al., 2023). Others suggest collecting attention variables independently from individual frame edits and applying them across frames (Ceylan et al., 2023; Wu et al., 2024); however, these independently generated attention variables often lack internal consistency.

In contrast, our recursive *gather* process ensures consistency within the attention group, which is especially crucial for long video generation, where maintaining coherence across thousands of frames is essential. Moreover, unlike previous methods that predominantly rely on self-attention, we also examine the significance of cross-attention layers, as highlighted in the ablation study.

Following the test-time adaptation process, each frame can be edited independently on separate GPUs during the spatiotemporal adaptation phase, significantly reducing the time required, particularly for long videos. We found that longer videos with more dynamics and scene changes benefit from a larger group size. In this work, we use a group size of 4 for all videos. The attention variable substitution process is performed throughout the entire denoising process, including the classifier-free guidance phase. The *gather* process is essential to the model's success. As shown in Fig. 13, for the same video, using the same random seed and editing instruction, attention gathering produces much more consistent group frames. Without the gathering process, although each frame in the group still follows the instruction, they exhibit different semantic editing directions. With the gathering process, the group maintains internal consistency, and the attention variables from it provide stable guidance for all video frames in the subsequent editing process.

## H   FURTHER IMPROVEMENT WITH BETTER ROOT FRAME

In our practice, we observed that a high-quality root frame pair generally leads to improved performance, as illustrated in Fig. 14. In Tab. 3, we show that performance can be further enhanced by incorporating an additional selector. It is important to note that neither a human selector nor an automatic selector was used during the comparison with baselines. By selecting the optimal frame based on editing quality, we ensure that the best possible results are achieved without requiring complex video-level adjustments. This streamlined approach significantly enhances the effectiveness of our

Table 3: **The selection strategy further improve the results.**

|  | Manuel | L1 | DINO | Random | No Test-time Adaptation |
|---|---|---|---|---|---|
| Frame-Acc ↑ | **0.891** | 0.882 | 0.887 | 0.873 | 0.871 |
| Tem-Con ↑ | **0.989** | 0.988 | 0.989 | 0.983 | 0.985 |
| Pixel-MSE ↓ | **0.0102** | 0.0107 | 0.0108 | 0.0111 | 0.0113 |

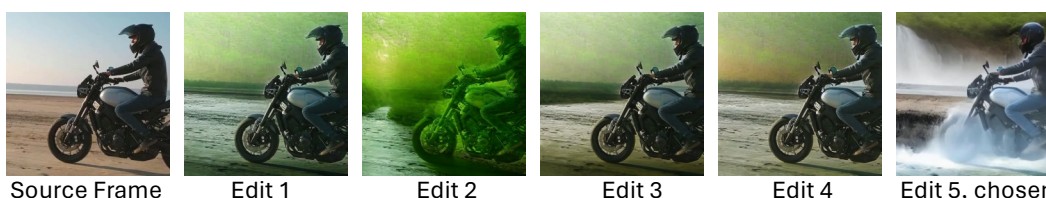

| Source Frame | Edit 1 | Edit 2 | Edit 3 | Edit 4 | Edit 5, chosen |

Figure 14: **Example of frame editing with different seeds.** Edited frames given the source frame on the left and editing instruction "Driving on a river in a forest"

method and addresses concerns related to frame selection, allowing for more consistent and visually appealing edits across the video.

# I   BROADER IMPACT

The advancements introduced by VIA have significant implications across various fields where video editing plays a crucial role. By enabling more precise, consistent, and efficient video editing, particularly for longer videos, VIA opens new possibilities for media production, education, and entertainment, among other domains. Here are some key areas of broader impact:

- **Media and Entertainment**: Our method allows filmmakers, content creators, and advertisers to produce higher-quality, longer-form content more efficiently. This could reduce production time and costs while enhancing the visual appeal and coherence of edited videos. Additionally, artists and creators can experiment with more complex and nuanced edits, fostering greater creative expression.

- **Education and Training**: Video is a key tool in educational content, and VIA can significantly improve the quality of instructional videos. Enhanced editing capabilities could lead to better engagement, clearer demonstrations, and more effective communication of ideas. For instance, complex concepts can be explained using tailored visual effects and transitions, making learning more accessible and intuitive.

- **Social Media and User-Generated Content**: As social media platforms increasingly rely on video content, our method can empower non-professional users to create polished, professional-quality videos. This could democratize access to high-end video editing, allowing users without technical expertise to achieve consistent, aesthetically pleasing results.

- **Advertising and Marketing**: In advertising, maintaining brand consistency across video content is critical. VIA's ability to ensure smooth transitions and coherent edits across frames can help marketers maintain the integrity of visual messaging over time, particularly in dynamic, minute-long commercials or social campaigns.

- **AI and Ethical Considerations**: While VIA improves video editing efficiency and quality, it also raises ethical concerns related to video manipulation. The ability to seamlessly edit long videos with high precision could potentially be misused for creating deepfakes or misleading media. As such, there is a need to implement ethical guidelines and detection mechanisms to ensure the responsible use of this technology. Additionally, transparency in editing processes and clear indicators of video manipulation will become increasingly important to prevent misinformation.

- **Environmental Impact**: By improving the efficiency of video editing, VIA reduces the computational resources required for long, complex video edits. This could lead to lower energy consumption, contributing to more environmentally sustainable video production workflows. Reducing the need for re-edits and long processing times could also have positive downstream effects on energy use in large-scale media production.

Overall, the broader impact of VIA extends beyond technical advancements, offering transformative potential across industries while also necessitating careful consideration of ethical and environmental responsibilities.

