# OpenReview forum: "VIA: Unified Spatiotemporal Video Adaptation for Global and Local Video Editing"
_ICLR.cc/2025/Conference — ICLR 2025 Conference Withdrawn Submission_

### Official Review · Reviewer_4Jor · 2024-10-27

**Soundness:** 2
**Presentation:** 2
**Contribution:** 2
**Rating:** 6
**Confidence:** 4

**Summary:**

The paper proposes a task-oriented framework called VIA. The framework combines test-time editing adaptation and spatial-temporal attention adaptation mechanisms to maintain both local frame consistency and global sequence coherence. VIA uses pre-trained image editing models and fine-tunes them using an enhanced dataset to improve their understanding and responsiveness to textual commands. Additionally, VIA employs aggregation and exchange processes to optimize cross-frame attention distribution, ensuring continuous and high-quality editing effects throughout the video. Experimental results show that VIA outperforms existing video editing methods in terms of accuracy, speed, and overall visual quality.

**Strengths:**

1. The paper presents a novel ramework for video editing that combines test-time editing adaptation and spatial-temporal attention adaptation mechanisms.

2. The paper presents a thorough literature review and compares the proposed method with several state-of-the-art methods. The experimental results demonstrate the effectiveness of the proposed method in terms of accuracy, speed, and overall visual quality.

3. The paper is well-written and easy to follow. The authors present the problem clearly and provide a detailed explanation of the proposed method.

**Weaknesses:**

I agree with the novelty of the method and the completeness of the work. However, from the video in the supplemental material, the edited results seem not good in terms of temporal consistency, and the video of comparison method is not provided. Could you provide more comparison videos?

**Questions:**

See Weaknesses.

---

### Official Review · Reviewer_ybnw · 2024-11-02

**Soundness:** 3
**Presentation:** 2
**Contribution:** 2
**Rating:** 3
**Confidence:** 4

**Summary:**

The authors propose VIA, to address the task of general video editing, esp. for maintaining long-term consistency in longer videos (1-minute). VIA uses a unified video editing framework built on pre-trained image editing approaches. It features test-time editing adaptation for local editing (instruction-following improvement) and spatial-temporal adaptation for long-term global editing (enhancing temporal coherence). The proposed method is compared with other related works and outperforms them in both quantitative and qualitative measurements.

**Strengths:**

- The proposed approach is technically sound and innovative. It is a good exploration to maintain temporal consistency in long-term video editing using cached attention variables from the individual frame editing process.
- The method is evaluated and compared with competitors through extensive experiments.
- The paper is well-motivated, as the existing methods struggle to preserve long-term content consistency due to a lack of effective propagation of editing information across frames.

**Weaknesses:**

- **Experiments**. The local editing results exhibit that VIA also edits the non-target regions/elements, which does not effectively support the authors' claims. For example, as shown in Fig.4, "Replace the animal to tiger" also edits the woman's face.
- **Experiments**. The ablation study experiments shown in Fig.8 (left) do not demonstrate significant performance improvement. In addition, "Our full model" even suffers from color-shifting issues across frames.
- **Unclear metric**. Pixel-MSE is not well explained, including its motivation and implementations. It also differs in scale from the original paper, i.e., Pix2Video. (Ceylan et al., ICCV'23)
- **Presentation**. The paper's presentation is somewhat concise, impacting readability, e.g., the proposed "progressive boundary integration", and those attention operations in "gather-and-swap". I recommend elaborating on key points to enhance logic and clarity, and making the paper more self-contained.
- **Presentation**. It would be beneficial to elaborate more on the baseline methods and analyze why they perform less effectively. This would provide clarity for the readers.

**Questions:**

- **Experiments**. The proposed method is built upon the image editing approach MGIE (Fu et al., ICLR'24). How does the underlying model impact performance, such as instruction following?

---

### Official Review · Reviewer_Xb9U · 2024-11-04

**Soundness:** 2
**Presentation:** 2
**Contribution:** 2
**Rating:** 5
**Confidence:** 4

**Summary:**

The paper presents a new framework, named VIA, which aims to perform text-guided video editing by addressing both local and global consistency challenges, particularly in minute-long videos. The paper proposes two main modules: (1) a test-time editing adaptation that fine-tunes a pre-trained image editing model for better alignment with text instructions, and (2) a spatiotemporal attention adaptation using a "gather-and-swap" strategy to maintain global consistency across frames. VIA outperforms existing methods in the aspects of fidelity, consistency, precision, and efficiency.

**Strengths:**

- Test-Time editing adaptation. The test-time adaptation mechanism allows the pre-trained image editing model to better align with text instructions. This ensures that edits are more faithful to user commands.

- Spatiotemporal attention adaptation. The presented "gather-and-swap" strategy for spatiotemporal attention adaptation effectively maintains global consistency across frames by sharing attention variables, leading to smoother transitions and coherent edits in the video sequence.

- Capability to handle long videos. The VIA framework can deal with longer video sequences, achieving results with temporal consistency and high quality.

**Weaknesses:**

## Unclear Writing and Presentation
- In the section on Progressive Boundary Integration, it appears that predefined masks are used to achieve precise and targeted edits. However, the methods used to obtain these masks, as well as their influence on the final performance, are not fully discussed.
- The details of the fine-tuning process in test-time adaptation are insufficient, making the work less reproducible. For instance, parameters such as learning rates, the number of training samples, and the number of iterations are not thoroughly explained.
## Insufficient Experiments
- The test-time adaptation seems generalizable to other similar methods. There should be more experiments to evaluate its general effectiveness, for example, by combining it with other state-of-the-art methods. The results of the full model in Figure 8 do not demonstrate obvious superiority over results without test-time adaptation. Additionally, the training samples used for test-time adaptation in Figure 8 are not provided, making the experimental results less reliable.
- In line 249, the paper mentions that Progressive Boundary Integration is different from other blending methods. However, the motivation and justifications behind this design, as well as comparisons with existing latent blending methods, are not provided, which makes the proposed module less convincing.
- The paper claims that the method can perform well with longer videos. However, only a few long video editing examples are presented in Figure 8.
- The paper mentions in line 861 that, to ensure fair comparison, the model is degraded to use only Spatiotemporal Adaptation during all evaluations. This experimental setting ignores the evaluation of other modules. A more appropriate comparison would be to equip other baseline models with the modules presented in this paper, examining their effectiveness. For example, apply Progressive Boundary Integration to all baseline models when conducting comparative studies.

**Questions:**

The same as listed in the "Weaknesses"

---

### Note · Authors · 2024-11-15

I have read and agree with the venue's withdrawal policy on behalf of myself and my co-authors.